# Molecular Role of Asn680Ser and Asp37Glu Missense Variants in Saudi Women with Female Infertility and Polycystic Ovarian Syndrome

Amal F. Alshammary [1,*][ID], Sarah F. Alsobaie [1][ID], Arwa A. Alageel [1], Fahad M. Aldakheel [1][ID], Sabah Ansar [1], Reem Alrashoudi [1], Raed Farzan [1][ID], Norah A. Alturki [1], Maysoon Abdulhadi Alhaizan [2], Johara Al-Mutawa [2] and Imran Ali Khan [1,*][ID]

1   Department of Clinical Laboratory Sciences, College of Applied Medical Sciences, King Saud University, Riyadh 11433, Saudi Arabia; salsobaie@ksu.edu.sa (S.F.A.); aaalageel@ksu.edu.sa (A.A.A.); faldakheel@ksu.edu.sa (F.M.A.); sansar@ksu.edu.sa (S.A.); ralrashoudi@ksu.edu.sa (R.A.); rfarzan@ksu.edu.sa (R.F.); nalturki@ksu.edu.sa (N.A.A.)
2   Department of Obstetrics and Gynecology, College of Medicine, King Saud University, Riyadh 11451, Saudi Arabia; malhaizan@ksu.edu.sa (M.A.A.); jalmutawa@ksu.edu.sa (J.A.-M.)
*   Correspondence: aalshammary@ksu.edu.sa (A.F.A.); imkhan@ksu.edu.sa (I.A.K.); Tel.: +966-501112806 (I.A.K.)

**Abstract:** Female infertility (FI) is a global health issue. Polycystic ovary syndrome (PCOS) is a common cause of FI. The renalase gene (*RNLS*) is associated with FI and other human diseases. Based on the documented missense variants, rs6166 and rs2296545 single-nucleotide polymorphisms (SNPs) were not identified in Saudi women with FI and PCOS. This study aimed to investigate the molecular role of the two SNPs in Saudi women with FI and PCOS. In this cross-sectional study, 96 healthy controls, 96 women with FI, and 96 women with PCOS were recruited. DNA was isolated, and polymerase chain reactions and Sanger sequencing analysis were performed using rs6166 and rs2296545 SNPs. The data obtained from the three groups were used to perform statistical analyses based on genotype, allele frequencies, regression models, and ANOVA analysis. Both rs6166 and rs2296545 had no role in FI or PCOS in Saudi women. A predicted reason for non-association in Saudi women could be the role of elderly women in the controls compared with women with FI and PCOS. Moreover, age, weight, and body mass index were higher in the control group than the FI and PCOS groups. In conclusion, rs6166 and rs2296545 SNPs were not associated with FI or PCOS in Saudi women.

**Keywords:** female infertility; polycystic ovary syndrome (PCOS); rs6166; rs2296545; FSHR; RNLS; missense variants; Saudi women

## 1. Introduction

Among human diseases, infertility is of major concern for married couples. The global definition of infertility is "the failure to obtain a clinical pregnancy after 365 days or more of frequent unprotected sexual intercourse." It is a common condition that affects both the male and female reproductive systems [1]. Currently, a substantial decline in the fertility rate has been observed globally [2]. The global prevalence of infertility in reproductive-aged women is one in every four couples in developing countries and one in every seven couples in the Western world [3]. Infertility has become a universal public health concern for married couples, with an estimated frequency of 3.5–26.4% globally, and 25% of married couples experience fertility issues during their lifetime of reproduction [4]. In primary infertility, a woman is clinically diagnosed as pregnant; however, in secondary infertility, the woman is unable to achieve a clinically diagnosed pregnancy. Globally, 10% of women experience primary or secondary infertility [5]. Approximately 48 million couples and 186 million individuals are affected [6]. The global infertility rates of women were

approximately 8%, 13–14%, and 18% in the ages of 19–26, 27–34, and 35–39 years, respectively [7]. The prevalence and causes of infertility in Saudi Arabia decreased from 7.3 to 3.03% during 1973–2010 and was 2.46% in 2020 (Figure 1). A study in Eastern Saudi Arabia confirmed that the prevalence of infertility was 18.93% [8]. A previous study in Northern Saudi Arabia confirmed that the prevalence of female infertility (FI) was 24.6% for ovulation defects, 3% for uterine fibroids, 3.2% for endometriosis, 6.7% for tubal adhesions, and 21.8% for polycystic ovaries. A combination of fibroids, PCOS (polycystic ovary syndrome), endometriosis, and reproductive tract infections is the major cause of FI [9]. Being precise, noninvasive, and cost-effective, ultrasonography is the primary method used to examine the pelvises of infertile women for evidence of the listed diseases. An advantage of employing ultrasound is that it provides accurate and detailed information that can be used to categorize probable problems of FI [10]. Infertility affects approximately 30% of males and 70% of females [11]. Various problems, such as ovulatory (25%), tubular (15%), and uterine (5%) factors, cause FI, and 25% of cases remain unexplained [12].

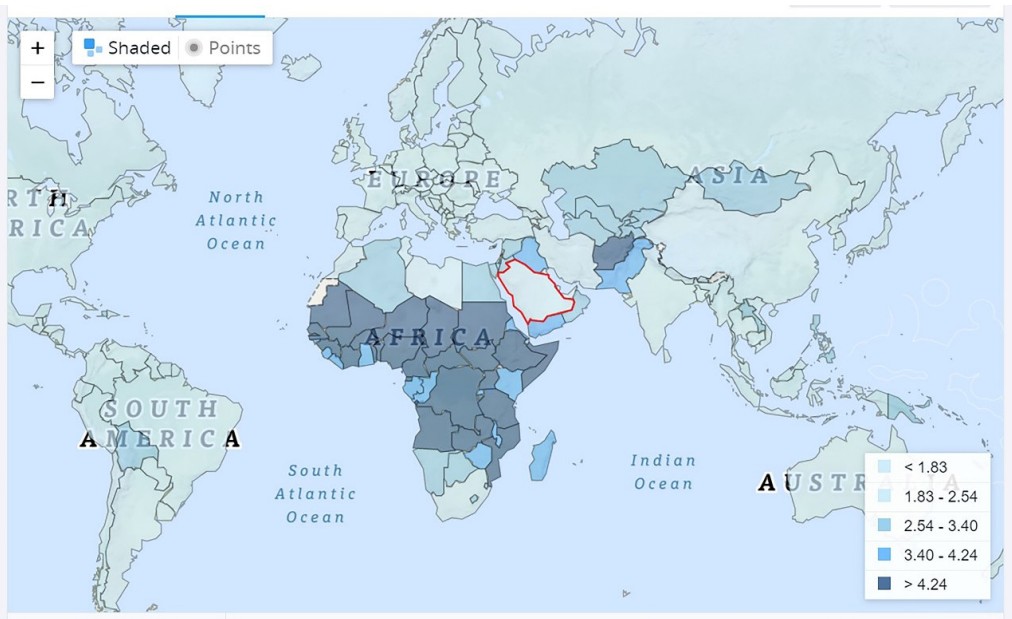

**Figure 1.** World Bank data specifies the total fertility rate of the global population from 1960 to 2020. The Kingdom of Saudi Arabia is indicated by a red line (source: www.dataworldbank.org; 10 January 2023).

PCOS is a major factor contributing to the development of FI [13]. PCOS is recorded as a chronic, heterozygous, and endocrine disorder in reproductive-aged women, which further manifests mainly as infertility, and then as menstrual dysfunction, acne, hirsutism, and obesity [14]. In 2003, PCOS was defined by the Rotterdam criteria as having at least a couple of classical features such as oligomenorrhoea, hyperandrogenism, and polycystic ovaries [15]. The National Institute of Health classified androgen excess and preferred abdominal fat deposition in women with PCOS in 1990 as being associated with an elevated risk of diabetes mellitus, dyslipidemia, and metabolic syndrome (MetS) [16]. A woman with PCOS mainly suffers from hormonal imbalance, insulin resistance (IR), and metabolic abnormalities, and subsequently develops FI, type 2 diabetes mellitus (T2DM), or cardiovascular disease (CVD), which affects the quality of the woman's life [17]. Based on different diagnostic criteria, the prevalence of PCOS was documented at 6–21% in global-wide ethnicities and regions [18]. IR and poor lipid metabolism have been linked to PCOS [19]. Abdominal adiposity, weight gain or obesity, and IR are commonly developing factors in women with PCOS, in addition to T2DM, CVD, MetS, and cancers [20]. The above complications are growing in the Saudi population, especially in the Saudi women [21–23]. The prevalence of gynecological disorders, such as PCOS and endometriosis, has increased

in Saudi Arabia due to changes in lifestyle factors [21]. Obesity and T2DM are growing rapidly within the Kingdom, and Saudi women are more likely to develop than the Saudi men to develop obesity [22,23]. Obese women frequently experience irregular menstruation with anovulation endometrial pathology with the frequent co-existence of PCOS [24]. The prevalence of both FI and PCOS is drastically increasing in Saudi women in Saudi Arabia. However, the accurate prevalence of PCOS was not documented in Saudi women. Based on an online survey study, the prevalence rate of PCOS was documented to be 16% in Saudi women [25].

Genetic factors strongly contribute to both FI and PCOS. However, genetic factors vary among twin-based, family-based, and case-control studies. Additionally, genetic anomalies such as single nucleotide polymorphisms (SNPs), single gene mutations, and chromosomal defects have been documented in infertile women [26].

Irregular menstruation can cause infertility in women [27], and PCOS is a common cause of abnormal menstruation [28]. Menstruation affects the secretion of sex hormones, and estrogen levels interact with IR. Elevated estrogen levels can affect the secretion of luteinizing hormone (LH) and lead to lower follicle-stimulating hormone (FSH) levels [29]. Many aspects of FI and PCOS defects are strongly influenced by genetic factors, and previous studies have attempted to identify the candidate genes; specific genes such as follicle-stimulating hormone receptors (FSHRs) can identify hormonal problems, which may eventually lead to either FI or PCOS [30,31]. The FSH is crucial for follicular growth and steroidogenesis. Female fertility relies on FSH receptor interactions, which are essential for follicular development and maturation [32]. The FSH is defined as an abnormal pattern of estrogen production during the first half of the menstrual cycle that can result in infertility or recurrent miscarriage [33]. The FSH is required for proper ovarian growth and function. The FSH induces ovulation by stimulating the growth and maturation of antral follicles in the ovaries [34]. Several parameters, including age, serum levels of the anti-Müllerian hormone, antral follicle, and number of women using transvaginal ultrasound, were ruled out to evaluate the ovarian reserve and possible predictors of the ovarian response toward patients for reproductive therapy. SNPs were additionally investigated to predict ovarian response, and the *FSHR* gene [35] was involved in IVF/ICSI cycles [36], FI [32], and PCOS [37]. The FSHR is a G protein-coupled receptor with ten exons, nine introns, and a promoter region on chromosome 2p21. The rs6166 SNP was Ser680Asn present in exon 10 with a variation in biological effects, and previous studies revealed that the rs6166 SNP was associated with an irregular menstrual cycle, ovarian hormone response, PCOS, FI, and the regulation of ovarian hyperstimulation [37].

We selected the rs2296545 SNP in the renalase (*RNLS*) gene because it is closely associated with FI [38], kidneys, T2DM, hypertension (HTN), and coronary artery disease [39]. Coincidentally, PCOS is linked to all the above-mentioned diseases [40], and these chronic diseases are common in the Saudi population [41,42]. RNLS is the first member of a novel family of monoamine oxidases that relies on the flavin adenine dinucleotide cofactor. Catecholamines in the bloodstream are metabolized through their expression in the kidneys and heart [43]. The *RNLS* gene has been traced to the 10q23.33 region of the human genome, and subsequent studies have shown that it is present in the reproductive system of women with infertility [38]. This gene has been identified in the oocytes, granulosa, interstitial, and luteal cells of the ovary [44]. The mechanism of renalase could influence FI via the cardiovascular effect of catecholamines, and both FI and cardiovascular risk are linked. A direct function was also linked between catecholamines and FI from the hypothalamic–pituitary–ovarian axis [38]. The rs2296545 missense SNP leads to an amino acid transition from glutamic to aspartic acid at position 37, which appears as Glu37Asp [43].

To date, no studies have been conducted on rs6166 and rs2296545 SNPs in the *FSHR* and *RNLS* genes in Saudi women with and without FI and PCOS. Both these SNPs and genes play separate roles in the Saudi population, and the present study examined rs6166 and rs2296545 SNPs in Saudi women diagnosed with FI and PCOS among the *FSHR* and *RNLS* genes.

## 2. Materials and Methods

### 2.1. Selection of Saudi Women

In this case-control study, 288 Saudi women were recruited from three different groups of subjects. The first group consisted of infertile women, the second group consisted of Saudi women diagnosed with PCOS, and the final group consisted of healthy controls. The study groups consisted of 96 samples from each of the three groups (infertile women, women with PCOS, and control subjects). This study was conducted at the outpatient clinic of the Obstetrics and Gynecology Department at King Khalid University Hospital and King Saud University (KSU) in the capital city of Saudi Arabia. Samples were collected over the entire year of 2021 for 12 months. The inclusion criteria for female infertility or infertility cases were based on Saudi women who did not conceive for a minimum of two and a maximum of three years, and family history was recorded. Saudi women with normal menstrual cycles, without any family history or ability to conceive were excluded from our female infertility study. PCOS cases were selected based on the presence of polycystic ovaries, Rotterdam criteria, and oligo-or anovulation in Saudi women. Women with PCOS who did not meet the Rotterdam criteria were excluded. Saudi women who met the Rotterdam criteria and had autoimmune diseases, ovarian lesions, and endometriosis were excluded as the controls. Normal healthy control Saudi women were based on regular menstruation, without any family/self-histories of FI/PCOS, non-Rotterdam criteria, conception, and a single ovary. The causes of infertility were investigated using a wide range of diagnostic tools, including hormonal and metabolic analyses, STD screening, hysterosalpingography, hysteroscopy, laparoscopy, the study of the partner's sperm, and an investigation of genetic immunological abnormalities. Women who participated in this study (n = 288) were in the age range of 18–40 years. Based on a questionnaire completed by Saudi participants, no patient was diagnosed with autoimmune, chronic, CVD, or any specific type of infection.

### 2.2. Ethical Approval

Ethical approval was received (E-19-4344 and E-20-5339) from the Institutional Review Board of the Medical College at KSU. All 288 Saudi women who participated in this study signed an informed consent form before enrollment. This study was performed according to the guidelines of the Declaration of Helsinki.

### 2.3. Anthropometric Measurements

Anthropometric measurements such as height were measured in centimeters and weight in kilograms, and body mass index (BMI) was calculated using the WHO standard, which is kilograms per meter squared ($kg/m^2$). Normal weight ($24.9\ kg/m^2$), overweight ($29.9\ kg/m^2$), obesity ($>30\ kg/m^2$), and severe/morbid obesity ($>35\ kg/m^2$) were the different types of obesity. Furthermore, the ages of all the women who participated were recorded in years based on their birth year, as well as different family histories.

### 2.4. Collection of Peripheral Blood Samples

In this study, we collected 6 mL of peripheral blood from 192 Saudi women (96 patients with PCOS and 96 healthy controls) and separated it into 4 mL of coagulated serum blood and 2 mL of EDTA blood. Serum samples were used for biochemical analysis, and EDTA blood was used for molecular analysis. From the 96 women with FI, 2 mL of peripheral blood was collected and stored in an EDTA vacutainer, which was later used for the extraction of genomic DNA.

### 2.5. Serum Sample Analysis

Fasting blood glucose (FBG), fasting insulin (FI), serum creatinine (SC)/creatinine, FSH, LH, thyroid-stimulating hormone (TSH), total testosterone (TT)/testosterone, and lipid profile parameters were defined as total cholesterol (TC), triglycerides (TGs), high-density lipoprotein cholesterol (HDLc) and low-density lipoprotein cholesterol (LDLc) levels and were measured with serum samples.

*2.6. Sanger Sequencing Method via Molecular Analysis*

Genomic DNA was extracted from 288 EDTA tubes of peripheral blood samples using a Qiagen kit according to the recommended protocol. Prior to amplification, DNA samples were stored at −80 °C and, the next day, the quality of 288 DNA samples was tested using a NanoDrop spectrophotometer. The DNA concentration was converted to 20 ng for the 288 DNA samples and used for further amplification analysis. Two different SNPs of specific genes, *RNLS* and *FSHR*, were genotyped using polymerase chain reaction (PCR) with accurate primers designed for rs6166 and rs2296545 SNPs. The total volume of PCR reaction mix was 50 µL with a master mix (10x Buffer, Mgcl$_2$, dNTPs, and Taq DNA polymerase) from a Qiagen kit, 10 pmoles of forward/reverse oligonucleotides, and 20 ng of genomic DNA. Finally, double-distilled water was used to bring up the volume to 50 µL. The protocol for PCR analysis included initial denaturation (95 °C-5 min), denaturation (95 °C-30 s), annealing (68 °C/62 °C for 30 s), extension (72 °C-45 s), and final extension (72 °C-5 min). The cycling condition was adjusted to 35 cycles and, after completion of the protocol, the PCR mix was held at 4 °C. The complete reaction time for both SNPs was 1.27–1.34 h. PCR products were run on a 2% agarose gel to confirm that 209 bp and 519 bp bands were present in the SNPs of *RNLS* and *FSHR*, respectively. The agarose gel was stained with ethidium bromide and documented using UVI gel doc. The PCR products were further purified and subjected to Sanger sequencing, as described in our previous studies. Using Big Dye for the PCR products and specific primers, the reaction was carried out using the Applied Biosystems Genetic Analyzer for the Sanger dideoxy chain termination method in the bidirectional process of high-throughput multicapillary sequencing. The sequenced file was received in the form of AB1, fasta, and PDF files, and the sequenced files were analyzed using the complete primer sequences of the rs2296545 and rs6166 SNPs examined in this study (Figure 2).

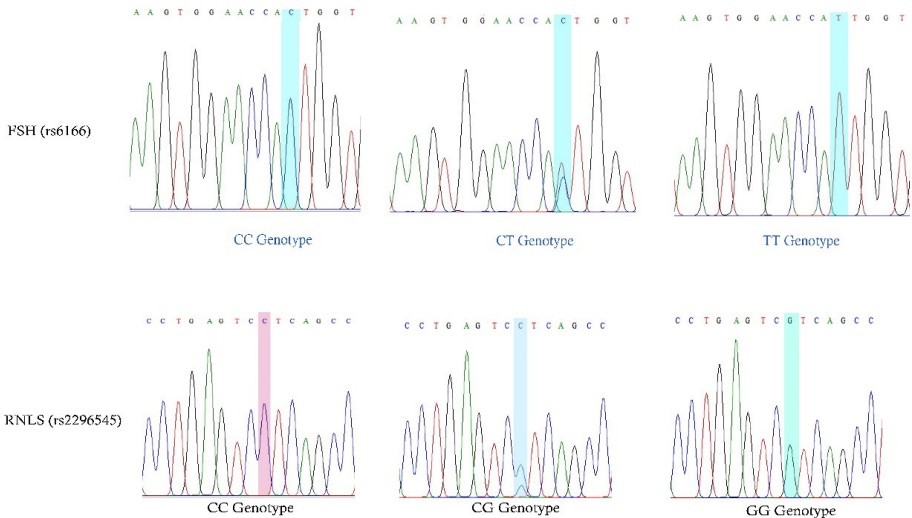

**Figure 2.** Sanger sequencing analysis revealed three different genotypes for both rs6166 and rs2996545 SNPs in *FSH* and *RNLS* genes.

*2.7. Statistical Analysis*

Anthropometric, biochemical, clinical, and molecular data were recorded in a Microsoft Excel document. Qualitative (mean and standard deviation) and quantitative (total number and percentage) data were analyzed. The baseline data presented were calculated using SPSS software in paired groups between the cases and controls (version 27.0). A Hardy–Weinberg Equilibrium (HWE) analysis is showed haplotype frequencies and linkage disequilibrium through the D' and r2 coefficients. Genotype analysis was carried out using a bivariate analysis of the association between FI and PCOS vs. SNPs studied with multiple genetic models, including allele frequencies, which were calculated using the odds ratio (OR), 95% confidence intervals (95% CI), and Pearson's $\chi^2$ and Fischer's exact tests with

SNPSTAT software. Multiple logistic regression analyses were performed for both SNPs in women with FI and PCOS using SPSS software. Analysis of variance (ANOVA) was performed using Jamovi software. Figure for the family histories of PCOS, and the control women were measured using Origin-Pro software. Using GraphPad software (version 9.5), Figure was drawn based on the BMI classification present in the three groups. The statistical significance was set at $p < 0.05$.

## 3. Results

### 3.1. Basic Details

In this study, 288 Saudi women classified into three groups were involved. The first group consisted of 96 subjects of female infertile women, confirmed by gynecologists based on personal history and laboratory analysis; the second group consisted of 96 women diagnosed with PCOS based on the Rotterdam criteria; and the final group of women (n = 96) were the normal control subjects without any history of FI or PCOS. In this study, the control women were commonly studied for FI and PCOS. The mean age, weight, and BMI of the control women were found to be higher than women with FI and PCOS because our main criteria in the selection of the control women were normal ovulation and Saudi nationality with a signed consent form.

### 3.2. Clinical Details of FI Subjects

The demographic details of the FI and control groups are shown in Table 1. The mean age of the control women and FI was found to be $31.39 \pm 6.70$ and $30.82 \pm 5.39$. When comparing FI with the control women, we found that age, weight, height, BMI, LH, and conception were higher ($p < 0.05$). Both FSH and TSH levels were higher in women with FI ($p < 0.0001$). All patients with FI were confirmed to be infertile ($p < 0.0001$), and 54.2% of the patients with FI had a family history of infertility ($p < 0.0001$).

**Table 1.** Demographic details present for female infertility (FI) and control subjects.

| Details | Female Infertility (n = 96) | Controls (n = 96) | *p*-Value |
|---|---|---|---|
| Age (Years) | $30.82 \pm 5.39$ | $31.39 \pm 6.70$ | 0.03 |
| Weight (kgs) | $73.92 \pm 11.33$ | $77.56 \pm 11.86$ | 0.03 |
| Height (cms) | $158.12 \pm 5.11$ | $159.02 \pm 6.88$ | 0.31 |
| BMI (kg/m$^2$) | $29.37 \pm 4.44$ | $30.68 \pm 4.53$ | 0.04 |
| FBG (mmol/L) | NA | $4.73 \pm 0.72$ | NA |
| FSH (IU/mL) | $7.33 \pm 0.78$ | $6.08 \pm 2.43$ | 0.0001 |
| LH (IU/mL) | $5.57 \pm 0.46$ | $6.95 \pm 2.36$ | <0.0001 |
| TSH (IU/mL) | $2.50 \pm 0.32$ | $2.01 \pm 0.64$ | <0.0001 |
| Conceived | 0 (0%) | 96 (100%) | <0.0001 |
| TC (mmol/L) | NA | $3.09 \pm 0.39$ | NA |
| TG (mmol/L) | NA | $1.59 \pm 0.89$ | NA |
| HDL-c (mmol/L) | NA | $0.48 \pm 0.15$ | NA |
| LDL-c (mmol/L) | NA | $3.20 \pm 0.56$ | NA |
| Infertility | 96 (100%) | 0 (0%) | <0.0001 |
| Family history of FI | 52 (54.2%) | 0 (0%) | <0.0001 |

NA = not analyzed; FI = female infertility.

### 3.3. Demographic Details of PCOS

Table 2 shows the demographic details of PCOS and the control women. The mean age of women with PCOS was $30.84 \pm 5.58$, which was low in comparison to the control women ($31.39 \pm 6.70$). However, weight was high in the control subjects. For BMI, women

with PCOS were found to be overweight (29.01 ± 4.76), and the control subjects were obese (30.68 ± 4.53). Women with PCOS showed elevated levels of FSH (6.91 ± 2.86), TT (1.89 ± 0.91), FBG (5.04 ± 0.88), FI (11.25 ± 6.36), serum creatinine (53.64 ± 13.33), TC (5.06 ± 1.07), HDL-c (0.63 ± 0.23), and LDLc (3.61 ± 0.92) parameters ($p < 0.05$). Other parameters, such as LH ($p = 0.27$), TSH ($p = 0.08$), and TG levels ($p = 0.16$), were found to be almost similar in both groups ($p > 0.05$). Family histories, such as T2DM, HTN, and a combination of T2DM and HTN, were found in 33.4% of the women with PCOS and 29.2% of the control subjects ($p = 0.74$). A family history of PCOS was confirmed in 29.2% of women with PCOS, and there was no history of PCOS in the control subjects ($p < 0.0001$).

**Table 2.** Demographic details of PCOS and control women.

| Details | PCOS (n = 96) | Controls (n = 96) | *p* Value |
|---|---|---|---|
| Age (Years) | 30.84 ± 5.58 | 31.39 ± 6.70 | 0.53 |
| Weight (kgs) | 73.61 ± 11.24 | 77.56 ± 11.86 | 0.01 |
| Height (cms) | 159.13 ± 5.10 | 159.02 ± 6.88 | 0.91 |
| BMI (kg/m$^2$) | 29.01 ± 4.76 | 30.68 ± 4.53 | 0.01 |
| FSH (IU/mL) | 6.91 ± 2.86 | 6.08 ± 2.43 | 0.03 |
| LH (IU/mL) | 7.54 ± 4.78 | 6.95 ± 2.36 | 0.27 |
| TSH (IU/mL) | 2.20 ± 0.86 | 2.01 ± 0.64 | 0.08 |
| Total testosterone (nmol/L) | 1.89 ± 0.91 | 0.89 ± 0.71 | <0.0001 |
| FBG (mmol/L) | 5.04 ± 0.88 | 4.73 ± 0.72 | 0.008 |
| Fasting insulin (μIU/mL) | 11.25 ± 6.36 | 9.05 ± 5.26 | 0.009 |
| Creatinine (mcmol/L) | 53.64 ± 13.33 | 45.11 ± 11.05 | 0.0001 |
| TC (mmol/L) | 5.06 ± 1.07 | 3.09 ± 0.39 | <0.0001 |
| TG (mmol/L) | 1.79 ± 1.09 | 1.59 ± 0.89 | 0.16 |
| HDL-c (mmol/L) | 0.63 ± 0.23 | 0.48 ± 0.15 | 0.0002 |
| LDL-c (mmol/L) | 3.61 ± 0.92 | 3.20 ± 0.56 | 0.0002 |
| Family history of PCOS | 28 (29.2%) | 0 (0%) | <0.0001 |
| Other family histories | 32 (33.4%) | 28 (29.2%) | 0.74 |

*3.4. Categorization of Women Based on Their Age*

Table 3 in this study has documented the complete details of all the women who were recruited. The participating women were categorized into three groups based on their ages, such as group I, which includes women aged between 18 and 20 years, group II, women between 21 and 30 years, and group III, women between 31 and 40 years. The control subjects consist of 7.3%, 36.5%, and 56.3% in all the three groups. LH (8.30 ± 2.63), TG (2.04 ± 1.33), and LDLc (3.31 ± 0.51) levels were found to be elevated in group I, whereas group II had FBG (4.83 ± 0.86), FI (8.97 ± 1.03), and TC (3.16 ± 0.35) levels that were elevated. Finally, in group III, weight (81.30 ± 12.64), height (159.65 ± 7.43), BMI (31.95 ± 4.88), FSH (6.58 ± 2.81), TSH (2.20 ± 0.31), and HDLc (0.49 ± 0.16) levels were found to be exceptionally high. There were no women present in the first group in FI subject; 56.3% of FI women were present in the second group, and height (157.78 ± 5.31), LH (5.58 ± 0.46), and TSH (2.52 ± 0.34) levels were substantially high. In the final group (group III), 47.9% of women were present, and weight (75.09 ± 10.01), BMI (29.73 ± 4.30), and FSH (7.34 ± 0.84) levels were elevated. PCOS women were divided into 5.2%, 37.5%, and 57.3% in three groups, and weight (77.40 ± 9.99), height (161.01 ± 6.16), BMI (29.80 ± 3.15), TSH (2.49 ± 0.90), and TG (2.25 ± 1.43) levels were found to be high in group I. FSH (7.32 ± 3.20), LH (8.40 ± 4.71), TC (5.24 ± 0.97), HDLc (0.68 ± 0.25), and LDLc (3.77 ± 0.98) levels were high in group II, and FBG (5.12 ± 0.94), fasting insulin

(11.37 ± 6.42), serum creatinine (54.42 ± 13.45), and total testosterone (1.94 ± 0.92) levels were high in group III.

**Table 3.** Categorization of the baseline details of the control subjects, FI and, PCOS based on three groups of age.

|  |  | Group 1 (n = 7) | Group 2 (n = 35) | Group 3 (n = 54) |
|---|---|---|---|---|
| Controls | Age (Years) | 19.00 ± 1.00 | 26.23 ± 2.71 | 36.48 ± 3.48 |
|  | Weight (Kgs) | 72.14 ± 9.71 | 73.13 ± 9.25 | 81.30 ± 12.64 |
|  | Height (cms) | 157.17 ± 5.77 | 158.04 ± 5.82 | 159.65 ± 7.43 |
|  | BMI (kg/m$^2$) | 29.11 ± 2.52 | 29.27 ± 3.73 | 31.95 ± 4.88 |
|  | FSH (IU/mL) | 6.41 ± 2.30 | 5.27 ± 1.57 | 6.58 ± 2.81 |
|  | LH (IU/mL) | 8.30 ± 2.63 | 6.47 ± 2.30 | 7.18 ± 2.33 |
|  | TSH (IU/mL) | 2.11 ± 0.30 | 2.05 ± 0.29 | 2.20 ± 0.31 |
|  | FBG (mmol/L) | 4.36 ± 0.40 | 4.83 ± 0.86 | 4.75 ± 0.64 |
|  | FI (µIU/mL) | 8.82 ± 0.64 | 8.97 ± 1.03 | 8.48 ± 1.16 |
|  | TC (mmol/L) | 3.12 ± 0.33 | 3.16 ± 0.35 | 3.07 ± 0.41 |
|  | TG (mmol/L) | 2.04 ± 1.33 | 1.60 ± 0.91 | 1.58 ± 0.83 |
|  | HDLc (mmol/L) | 0.45 ± 0.13 | 0.47 ± 0.14 | 0.49 ± 0.16 |
|  | LDLc (mmol/L) | 3.31 ± 0.51 | 3.23 ± 0.50 | 3.16 ± 0.62 |
|  |  | Group 1 (n = 0) | Group 2 (n = 46) | Group 3 (n = 50) |
| Female Infertility | Age (Years) | 0.00 ± 0.00 | 26.46 ± 3.05 | 34.84 ± 3.67 |
|  | Weight (Kgs) | 0.00 ± 0.00 | 72.46 ± 12.61 | 75.09 ± 10.01 |
|  | Height (cms) | 0.00 ± 0.00 | 157.78 ± 5.31 | 157.66 ± 4.97 |
|  | BMI (kg/m$^2$) | 0.00 ± 0.00 | 28.97 ± 4.59 | 29.73 ± 4.30 |
|  | FSH (IU/mL) | 0.00 ± 0.00 | 7.31 ± 0.73 | 7.34 ± 0.84 |
|  | LH (IU/mL) | 0.00 ± 0.00 | 5.58 ± 0.46 | 5.57 ± 0.47 |
|  | TSH (IU/mL) | 0.00 ± 0.00 | 2.52 ± 0.34 | 2.48 ± 0.30 |
|  |  | Group 1 (n = 5) | Group 2 (n = 36) | Group 3 (n = 55) |
| PCOS | Age (Years) | 18.40 ± 0.55 | 26.53 ± 2.44 | 34.80 ± 2.85 |
|  | Weight (Kgs) | 77.40 ± 9.99 | 74.49 ± 10.25 | 72.69 ± 12.01 |
|  | Height (cms) | 161.01 ± 6.16 | 159.08 ± 4.98 | 158.99 ± 5.15 |
|  | BMI (kg/m$^2$) | 29.80 ± 3.15 | 29.51 ± 4.49 | 28.61 ± 5.07 |
|  | FBG (mmol/L) | 4.85 ± 0.99 | 4.95 ± 0.78 | 5.12 ± 0.94 |
|  | FI (µIU/mL) | 11.36 ± 6.67 | 11.07 ± 6.41 | 11.37 ± 6.42 |
|  | Creatinine (mcmol/L) | 45.60 ± 12.24 | 53.56 ± 13.27 | 54.42 ± 13.45 |
|  | FSH (IU/mL) | 6.88 ± 1.44 | 7.32 ± 3.20 | 6.64 ± 2.73 |
|  | LH (IU/mL) | 5.62 ± 4.54 | 8.40 ± 4.71 | 7.16 ± 4.84 |
|  | TSH (IU/mL) | 2.49 ± 0.90 | 2.06 ± 0.79 | 2.26 ± 0.92 |
|  | TT (nmol/L) | 1.83 ± 0.43 | 1.82 ± 0.94 | 1.94 ± 0.92 |
|  | TC (mmol/L) | 4.97 ± 1.19 | 5.24 ± 0.97 | 4.95 ± 1.13 |
|  | TG (mmol/L) | 2.25 ± 1.43 | 1.75 ± 0.99 | 1.78 ± 1.12 |
|  | HDLc (mmol/L) | 0.54 ± 0.20 | 0.68 ± 0.25 | 0.61 ± 0.22 |
|  | LDLc (mmol/L) | 3.42 ± 0.75 | 3.77 ± 0.98 | 3.53 ± 0.89 |

Group 1 is between 18 and 20 years of age, group 2 is between 21 and 30 years, and group 3 is between 31 and 40 years of age for the women involved in this study.

### 3.5. HWE Analysis Studied between rs6166 and rs2296545 SNPs

The purpose of the HWE analysis was to rectify genotyping errors and estimate the presence of total heterozygous and homozygous variants. In this study, we performed HWE analysis of rs6166 and rs2296545 SNPs in FI, PCOS, and the healthy controls. The *p*-value of the genotype distribution in each group is shown in Table 4. All three groups were found to be in HWE equilibrium ($p < 0.05$). The HWE for each group was calculated using the chi-square test.

**Table 4.** HWE analysis studied between rs6166 and rs2296545 SNPs in FI and PCOS.

| SNP ID | Total Samples | Controls (n = 96) | $\chi^2$ | MAF | *p* | PCOS (n = 96) | $\chi^2$ | MAF | *p* | FI (n = 96) | $\chi^2$ | MAF | *p* |
|---|---|---|---|---|---|---|---|---|---|---|---|---|---|
| rs6166 | 96 (100%) | 86/08/02 | 8.01 | 0.06 | 0.004 | 76/16/04 | 5.44 | 0.13 | 0.01 | 82/10/04 | 14.37 | 0.09 | 0.0001 |
| rs2296545 | 96 (100%) | 63/23/10 | 9.27 | 0.22 | 0.002 | 59/25/12 | 9.52 | 0.26 | 0.002 | 78/14/04 | 7.59 | 0.11 | 0.005 |

### 3.6. Genotyping Analysis of Asn680Ser Variants in FI and PCOS

The CC, CT, and TT genotype frequencies were 85.4%, 10.4%, and 4.2% in FI cases; 79.2%, 16.6%, and 4.2% in women with PCOS; and 89.6%, 8.3%, and 2.1% in the control women, respectively. Normal alleles were found to be high in the controls (93.75%), FI cases (90.62%), and women with PCOS (87.50%), whereas mutant alleles were found to be high in women with PCOS (12.50%), FI cases (9.38%), and the control subjects (6.25%). The statistics and frequencies of rs6166 (Asn680Ser) are presented in Table 5. None of the genotypes (CT vs. CC: OR- OR-1.31 (95% CI (0.49–3.48)); $p = 0.58$, TT vs. CC: OR-2.09 (95% CI (0.37–11.76)); $p = 0.39$), genetic models (CT + TT vs. CC: OR-1.46 (95% CI (0.61–3.49)); $p = 0.38$, TT + CC vs. CT: OR-0.97 (95% CI (0.38–2.46)); $p = 0.96$ and CC + CT vs. TT: OR-0.48 (95% CI (0.08–2.73)); $p = 0.40$), and allele frequencies (T vs. C: OR-1.55 (95% CI (0.72–3.31)); $p = 0.25$) were found to be non-significant in both FI, as well as in women with PCOS (CT vs. CC: OR-2.26 (95% CI (0.91–5.58)); $p = 0.07$, TT vs. CC: OR-1.31 (95% CI (0.27–4.68)); $p = 0.86$), genetic models (CT + TT vs. CC: OR-2.26 (95% CI (0.99–5.13)); $p = 0.04$, TT + CC vs. CT: OR-0.58 (95% CI (0.24–1.35)); $p = 0.20$ and CC + CT vs. TT: OR-1.00 (95% CI (0.24–4.11)); $p = 0.99$), and allele frequencies (T vs. C: OR-1.38 (95% CI (0.72–2.63)); $p = 0.32$).

**Table 5.** Genotype distribution and allele frequencies in the rs6166/rs2296525 SNPs compared with the controls versus women with FI and PCOS.

| Gene (rsnumber) | Genotypes | FI Cases (n = 96) | Controls (n = 96) | OR (95%CI) *p*-Value | PCOS (n = 96) | OR (95%CI) *p*-Value |
|---|---|---|---|---|---|---|
| FSH (rs6166) | CC | 82 (85.4%) | 86 (89.6%) | Reference | 76 (79.2%) | Reference |
| | CT | 10 (10.4%) | 08 (8.3%) | OR-1.31 [95% CI (0.49–3.48)]; $p = 0.58$ | 16 (16.6%) | OR-2.26 [95% CI (0.91–5.58)]; $p = 0.07$ |
| | TT | 04 (4.2%) | 02 (2.1%) | OR-2.09 [95% CI (0.37–11.76)]; $p = 0.39$ | 04 (4.2%) | OR-1.31 [95% CI (0.27–4.68)]; $p = 0.86$ |
| | CT + TT vs. CC | 14 (14.6%) | 10 (10.4%) | OR-1.46 [95% CI (0.61–3.49)]; $p = 0.38$ | 20 (20.8%) | OR-2.26 [95% CI (0.99–5.13)]; $p = 0.04$ |
| | TT + CC vs. CT | 86 (89.6%) | 88 (91.7%) | OR-0.97 [95% CI (0.38–2.46)]; $p = 0.96$ | 80 (83.4%) | OR-0.58 [95% CI (0.24–1.35)]; $p = 0.20$ |
| | CC + CT vs. TT | 92 (95.8%) | 94 (97.9%) | OR-0.48 [95% CI (0.08–2.73)]; $p = 0.40$ | 92 (95.8%) | OR-1.00 [95% CI (0.24–4.11)]; $p = 0.99$ |
| | C allele | 174 (90.62%) | 180 (93.75%) | Reference | 168 (87.5%) | Reference |
| | T allele | 18 (9.38%) | 12 (6.25%) | OR-1.55 [95% CI (0.72–3.31)]; $p = 0.25$ | 24 (12.5%) | OR-1.38 [95% CI (0.72–2.63)]; $p = 0.32$ |
| RNLS (rs2296545) | CC | 78 (81.3%) | 63 (65.6%) | Reference | 59 (61.5%) | Reference |
| | CG | 14 (14.6%) | 23 (24.0%) | OR-0.49 [95% CI (0.23–1.03)]; $p = 0.58$ | 25 (26.0%) | OR-1.16 [95% CI (0.59–2.26)]; $p = 0.66$ |
| | GG | 04 (4.2%) | 10 (10.4%) | OR-0.32 [95% CI (0.09–1.07)]; $p = 0.05$ | 12 (12.5%) | OR-1.28 [95% CI (0.51–3.18)]; $p = 0.59$ |
| | CG + GG vs. CC | 18 (18.2%) | 33 (34.4%) | OR-0.44 [95% CI (0.22–0.85)]; $p = 0.58$ | 37 (38.5%) | OR-1.19 [95% CI (0.66–2.15)]; $p = 0.54$ |

**Table 5.** *Cont.*

| Gene (rsnumber) | Genotypes | FI Cases (n = 96) | Controls (n = 96) | OR (95%CI) *p*-Value | PCOS (n = 96) | OR (95%CI) *p*-Value |
|---|---|---|---|---|---|---|
| RNLS (rs2296545) | GG + CC vs. CG | 82 (85.5%) | 73 (76.0%) | OR-1.85 [95% CI (0.88–3.85)]; *p* = 0.09 | 71 (74.0%) | OR-0.89 [95% CI (0.46–1.72)]; *p* = 0.73 |
| | CC + CG vs. GG | 92 (95.8%) | 86 (89.6%) | OR-2.67 [95% CI (0.80–8.84)]; *p* = 0.09 | 84 (87.5%) | OR-0.30 [95% CI (0.09–0.98)]; *p* = 0.03 |
| | C allele | 170 (88.54%) | 149 (77.6%) | Reference | 143 (74.48%) | Reference |
| | G allele | 22 (11.46%) | 43 (22.4%) | OR-0.44 [95% CI (0.25–0.78)]; *p* = 0.004 | 49 (25.52%) | OR-1.18 [95% CI (0.74–1.89)]; *p* = 0.47 |

*3.7. Molecular Analysis of Asp37Glu Variants in FI and PCOS*

Table 5 also demonstrates the genotype and allele frequencies of the rs2296545 (Asp37Glu) SNP in FI cases, women with PCOS, and the control subjects. The C allele frequencies were 88.54% in FI cases, 77.60% in women with PCOS, and 74.48% % in the control subjects. The G allele was found in 11.46% of FI cases, 25.52% of PCOS cases, and 22.40% in the control subjects. The FI cases had CC, CG, and GG genotype frequencies of 81.3%, 14.6%, and 4.2%. The women with PCOS had 61.5%, 26.0%, 12.5%, and 65.6%, 24.0%, and 10.4% in the control subjects, which documented the genotype frequencies present in Asp37Glu variants in all three groups. The statistical analysis was performed in the genotype analysis (CG vs. CC: OR-0.49 (95% CI (0.23–1.03)); *p* = 0.58, GG vs. CC: OR-0.32 (95% CI (0.09–1.07)); *p* = 0.05), genetic models (CG + GG vs. CC: OR-0.44 (95% CI (0.22–0.85)); *p* = 0.58, GG + CC vs. CG: OR-1.85 (95% CI (0.88–3.85)); *p* = 0.09 and CC + CG vs. GG: OR-2.67 (95%CI (0.80–8.84)); *p* = 0.09), and allele frequencies (G vs. C: OR-0.44 (95% CI (0.25–0.78)); *p* = 0.004) showed no association between FI cases and women with PCOS in the genotype analysis (CG vs. CC: OR-1.16 (95% CI (0.59–2.26)); *p* = 0.66, GG vs. CC: OR-1.28 (95% CI (0.51–3.18)); *p* = 0.59), genetic models (CG + GG vs. CC: OR-1.19 (95% CI (0.66–2.15)); *p* = 0.54, GG + CC vs. CG: OR-0.89 (95% CI (0.46–1.72)); *p* = 0.73 and CC + CG vs. GG: OR-0.30 (95% CI (0.09–0.98)); *p* = 0.03), and allele frequencies (G vs. C: OR-1.18 (95% CI (0.74–1.89)); *p* = 0.47).

*3.8. Multiple Logistic Regression Analysis between rs6166 and rs2296545 SNPs in FI and PCOS Subjects*

Table 6 in this study has described the multiple logistic regression analysis of FI subjects with rs6166 and rs2296545 SNPs in *FSHR* and *RNLS*. In rs6166SNP, TT was used as a reference to study the remaining genotypes (CC and CT), along with covariates including age, weight, BMI, FSH, LH, TSH, and rs2296545 SNP. No association was found for any of the parameters studied in either the CC or CT genotypes (*p* > 0.05). Table 7 in this study defined the relationship between rs6166 and rs2296545 SNPs in women with PCOS. In this study, the covariates of PCOS included age, weight, and BMI, and FSH, LH, TSH, FBG, FI, creatinine, total testosterone, TC, TG, HDLc, and LDLc levels. Both CC and CT genotypes were studied using the TT genotype as a reference, and rs2296545 SNP was involved. A multiple logistic regression analysis confirmed that there was no association with any of the covariates for either the CC or CT genotypes (*p* > 0.05).

**Table 6.** Logistic regression analysis studies of rs6166, rs2296545 SNPs, and FI subjects.

| rs6166 [a] CC Genotypes | B | SE | Wald | df | Significance | Exp(B) | 95% Confidence Intervals for Exp(B) | |
|---|---|---|---|---|---|---|---|---|
| | | | | | | | Lower Bound | Upper Bound |
| Age (Years) | −0.176 | 0.147 | 1.423 | 1 | 0.233 | 0.839 | 0.629 | 1.120 |
| Weight (kgs) | −0.541 | 0.461 | 1.376 | 1 | 0.241 | 0.582 | 0.236 | 1.438 |
| BMI (kg/m$^2$) | −1.229 | 1.158 | 1.126 | 1 | 0.289 | 0.293 | 0.030 | 2.831 |
| FSH (IU/mL) | 0.579 | 0.718 | 0.651 | 1 | 0.420 | 1.784 | 0.437 | 7.283 |
| LH (IU/mL) | −0.804 | 1.185 | 0.460 | 1 | 0.498 | 0.448 | 0.044 | 4.566 |

**Table 6.** *Cont.*

| rs6166 [a] CC Genotypes | B | SE | Wald | df | Significance | Exp(B) | 95% Confidence Intervals for Exp(B) | |
|---|---|---|---|---|---|---|---|---|
| | | | | | | | Lower Bound | Upper Bound |
| TSH (IU/mL) | −0.173 | 1.801 | 0.009 | 1 | 0.923 | 0.841 | 0.025 | 28.688 |
| [rs2296545 = CC] | −18.381 | 1.992 | 85.143 | 1 | <0.001 | $1.040 \times 10^{-8}$ | $2.097 \times 10^{-10}$ | $5.162 \times 10^{-7}$ |
| [rs2296545 = CG] | −18.904 | 1.495 | 159.916 | 1 | <0.001 | $6.165 \times 10^{-9}$ | $3.29 \times 10^{-10}$ | $1.155 \times 10^{-7}$ |
| [rs2296545 = GG] | 0 [b] | - | - | 0 | - | - | - | - |

| rs6166 [a] CT Genotypes | B | SE | Wald | df | Significance | Exp(B) | 95% Confidence Intervals for Exp(B) | |
|---|---|---|---|---|---|---|---|---|
| | | | | | | | Lower Bound | Upper Bound |
| Age (Years) | −0.155 | 0.159 | 0.947 | 1 | 0.330 | 0.856 | 0.627 | 1.170 |
| Weight (kgs) | −0.402 | 0.509 | 0.624 | 1 | 0.430 | 0.669 | 0.247 | 1.814 |
| BMI (kg/m$^2$) | −0.792 | 1.279 | 0.383 | 1 | 0.536 | 0.453 | 0.037 | 5.558 |
| FSH (IU/mL) | 0.631 | 0.809 | 0.609 | 1 | 0.435 | 1.880 | 0.385 | 9.179 |
| LH (IU/mL) | −0.679 | 1.355 | 0.251 | 1 | 0.616 | 0.507 | 0.036 | 7.222 |
| TSH (IU/mL) | 0.435 | 2.045 | 0.045 | 1 | 0.831 | 1.546 | 0.028 | 85.004 |
| [rs2296545 = CC] | −18.043 | 1.714 | 110.836 | 1 | <0.001 | $1.459 \times 10^{-8}$ | $5.074 \times 10^{-10}$ | $4.197 \times 10^{-7}$ |
| [rs2296545 = CG] | −19.535 | 0.000 | - | 1 | - | $3.283 \times 10^{-9}$ | $3.283 \times 10^{-9}$ | $3.283 \times 10^{-9}$ |
| [rs2296545 = GG] | 0 [b] | - | - | 0 | - | - | - | - |

[a] Reference category is TT genotype and [b] this parameter is set to zero because it is redundant.

**Table 7.** Multiple logistic regression analysis studies of rs6166, rs2296545 SNPs, and PCOS subjects.

| rs6166 [a] CC Genotypes | B | SE | Wald | df | Significance | Exp(B) | 95% Confidence Intervals for Exp(B) | |
|---|---|---|---|---|---|---|---|---|
| | | | | | | | Lower Bound | Upper Bound |
| Age (Years) | 1.951 | 1.941 | 1.010 | 1 | 0.315 | 7.034 | 0.157 | 315.970 |
| Weight (kgs) | 1.532 | 1.437 | 1.136 | 1 | 0.286 | 4.629 | 0.277 | 77.450 |
| BMI (kg/m2) | −5.383 | 3.953 | 1.854 | 1 | 0.173 | 0.005 | $1.985 \times 10^{-6}$ | 10.642 |
| FSH (IU/mL) | −0.453 | 1.340 | 0.114 | 1 | 0.735 | 0.636 | 0.046 | 8.783 |
| LH (IU/mL) | 1.991 | 2.748 | 0.525 | 1 | 0.469 | 7.321 | 0.034 | 1599.193 |
| TSH (IU/mL) | 5.848 | 7.281 | 0.645 | 1 | 0.422 | 346.369 | 0.000 | 546405038.0 |
| FBG (mmol/L) | −1.427 | 15.256 | 0.009 | 1 | 0.925 | 0.240 | $2.477 \times 10^{-14}$ | $2.325 \times 10^{12}$ |
| FI (µIU/mL) | 1.591 | 3.541 | 0.202 | 1 | 0.653 | 4.909 | 0.005 | 5075.065 |
| Creatinine (mcmol/L) | 0.347 | 0.348 | 0.996 | 1 | 0.318 | 1.415 | 0.716 | 2.796 |
| Testosterone | −2.626 | 8.240 | 0.102 | 1 | 0.750 | 0.072 | $7.004 \times 10^{-9}$ | 747874.150 |
| TC (mmol/L) | 1418.3 | 70.601 | 403.558 | 1 | <0.001 | .[b] | .[b] | .[b] |
| TG (mmol/L) | −627.01 | 37.153 | 284.81 | 1 | <0.001 | $4.971 \times 10^{-2}$ | $1.180 \times 10^{-304}$ | $2.094 \times 10^{-241}$ |
| HDL-c (mmol/L) | −1381.3 | 90.125 | 234.89 | 1 | <0.001 | 0.000 | 0.000 | 0.000 |
| LDL-c (mmol/L) | −1411.4 | 68.867 | 420.030 | 1 | <0.001 | 0.000 | 0.000 | 0.000 |
| [rs2296545 = CC] | −25.889 | 68.636 | 0.142 | 1 | 0.706 | $5.709 \times 10^{-1}$ | $2.154 \times 10^{-70}$ | $1.513 \times 10^{47}$ |
| [rs2296545 = CG] | −15.909 | 79.189 | 0.040 | 1 | 0.841 | $1.233 \times 10^{-7}$ | $4.841 \times 10^{-75}$ | $3.139 \times 10^{60}$ |
| [rs2296545 = GG] | 0 [b] | - | - | 0[c] | - | - | - | - |

**Table 7.** *Cont.*

| rs6166 [a] CT Genotypes | B | SE | Wald | df | Significance | Exp(B) | 95% Confidence Intervals for Exp(B) | |
|---|---|---|---|---|---|---|---|---|
| | | | | | | | Lower Bound | Upper Bound |
| Age (Years) | 1.959 | 1.942 | 1.018 | 1 | 0.313 | 7.092 | 0.158 | 319.030 |
| Weight (kgs) | 1.629 | 1.438 | 1.282 | 1 | 0.258 | 5.097 | 0.304 | 85.453 |
| BMI (kg/m$^2$) | −5.683 | 3.955 | 2.065 | 1 | 0.151 | 0.003 | $1.463 \times 10^{-6}$ | 7.915 |
| FSH (IU/mL) | −0.483 | 1.344 | 0.129 | 1 | 0.719 | 0.617 | 0.044 | 8.600 |
| LH (IU/mL) | 1.995 | 2.749 | 0.527 | 1 | 0.468 | 7.354 | 0.034 | 1607.602 |
| TSH (IU/mL) | 5.415 | 7.292 | 0.552 | 1 | 0.458 | 224.835 | 0.000 | 361838.5 |
| FBG (mmol/L) | −1.417 | 15.262 | 0.009 | 1 | 0.926 | 0.242 | $2.474 \times 10^{-14}$ | $2.373 \times 10^{12}$ |
| FI (µIU/mL) | 1.647 | 3.542 | 0.216 | 1 | 0.642 | 5.191 | 0.005 | 5374.834 |
| Creatinine (mcmol/L) | 0.371 | 0.348 | 1.137 | 1 | 0.286 | 1.449 | 0.733 | 2.865 |
| Testosterone | −2.008 | 8.245 | 0.059 | 1 | 0.808 | 0.134 | $1.288 \times 10^{-8}$ | 1400705.306 |
| TC (mmol/L) | 1416.4 | 15.613 | 8230.1 | 1 | 0.000 | .[b] | .[b] | .[b] |
| TG (mmol/L) | −626.06 | 20.481 | 934.43 | 1 | <0.001 | $1.269 \times 10^{-3}$ | $4.681 \times 10^{-290}$ | $3.443 \times 10^{-255}$ |
| HDL-c (mmol/L) | −1378.9 | 59.220 | 542.20 | 1 | <0.001 | 0.000 | 0.000 | 0.000 |
| LDL-c (mmol/L) | −1409.6 | 0.000 | | 1 | | 0.000 | 0.000 | 0.000 |
| [rs2296545 = CC] | −26.175 | 68.638 | 0.145 | 1 | 0.703 | $4.288 \times 10^{-1}$ | $1.611 \times 10^{-70}$ | $1.141 \times 10^{47}$ |
| [rs2296545 = CG] | −17.180 | 79.196 | 0.047 | 1 | 0.828 | $3.456 \times 10^{-8}$ | $1.340 \times 10^{-75}$ | $8.915 \times 10^{59}$ |
| [rs2296545 = GG] | 0 [b] | - | - | 0 [c] | - | - | - | - |

[a] Reference category is TT; [b] floating point overflow occurred while computing this statistic. Its value is, therefore, set to system missing. [c] This parameter is set to zero because it is redundant.

### 3.9. ANOVA Analysis in Women with FI and PCOS with rs6166 and rs2296545 SNPs

ANOVA analysis was performed between rs6166 and rs2296545 SNPs in FI subjects and mentioned in Table 8. Age, weight, and BMI, and TSH, LH, and FSH levels were used as covariates in this study. Both FSH (7.37 ± 0.31) and TSH (2.54 ± 0.36) covariates had elevated levels present in CT genotypes, and age (33.00 ± 7.79), weight (81.75 ± 9.54), BMI (31.70 ± 3.64), and LH (5.68 ± 0.48) were found to be high in TT genotypes in FI cases. None of the covariates were associated with rs6166 in subjects with FI ($p > 0.05$). Among rs2296545 SNP, weight (75.24 ± 10.53), BMI (29.97 ± 3.94), and TSH (2.52 ± 0.31) levels were found to be high in CC genotypes, and age (31.50 ± 4.53), FSH (7.72 ± 0.46), and LH (5.65 ± 0.44) levels were found to be high in GG genotypes, and rs2296545 SNP was not associated with any of the covariates in FI subjects ($p > 0.05$). ANOVA was also performed on women with PCOS using the rs6166 and rs2296545 SNPs. Age, weight, and BMI, and FBG, FI, creatinine, FSH, LH, TSH, testosterone, TC, TG, and HDL-C levels were considered covariates, as shown in Table 9. In rs6166 SNP, TC (5.10 ± 1.08), TG (1.82 ± 1.08), and LDLc (3.64 ± 0.91) levels were found to be elevated in the CC genotype. Age (31.31 ± 4.06), FBG (5.36 ± 1.13), FI (13.47 ± 8.73), creatinine (57.75 ± 21.85), LH (7.74 ± 3.23), testosterone (2.29 ± 1.12), and HDLc (0.67 ± 0.27) levels were elevated in CG genotypes, and weight (75.95 ± 14.13), BMI (31.08 ± 5.47), FSH (7.88 ± 4.23), and TSH (2.43 ± 0.56) levels were high in the GG genotypes. There was no positive association with any of the covariates in rs6166 SNP ($p > 0.05$). Regarding rs2296545 SNP, weight (74.24 ± 10.67), BMI (29.29 ± 4.46), creatinine (54.69 ± 14.64), FSH (7.24 ± 3.43), LH (8.38 ± 5.22), TSH (2.23 ± 0.74), TC (5.08 ± 1.12), and TG (1.93 ± 1.25) levels were found to be elevated in tge CC genotypes, and LDLc (3.70 ± 0.85) levels were found to be elevated in the CG genotype and, in GG genotypes, age (31.00 ± 5.67), FBG (5.61 ± 1.15), FI (13.42 ± 8.18), testosterone (2.21 ± 1.36), and HDLc

(0.69 ± 0.23) levels were elevated. There was no association between rs2296545 and the covariates present in any of the women with PCOS ($p > 0.05$).

**Table 8.** ANOVA analysis studies of FI subjects, rs6166, and rs2296545 SNPs.

| Covariates | rs6166 | | | | rs2296545 | | | |
|---|---|---|---|---|---|---|---|---|
| | CC = 82 | CT = 10 | TT = 04 | *p*-Value | CC = 63 | CG = 23 | GG = 10 | *p*-Value |
| Age (Years) | 30.68 ± 5.47 | 31.10 ± 3.84 | 33.00 ± 7.79 | 0.69 | 30.73 ± 5.28 | 30.78 ± 6.19 | 31.50 ± 4.53 | 0.96 |
| Weight (kgs) | 73.48 ± 11.22 | 74.42 ± 12.69 | 81.75 ± 9.54 | 0.36 | 75.24 ± 10.53 | 72.09 ± 12.43 | 69.82 ± 13.15 | 0.25 |
| BMI (kg/m$^2$) | 29.17 ± 4.43 | 30.02 ± 4.83 | 31.70 ± 3.64 | 0.48 | 29.97 ± 3.94 | 28.15 ± 5.12 | 28.35 ± 5.39 | 0.18 |
| FSH (IU/mL) | 7.34 ± 0.83 | 7.37 ± 0.31 | 6.95 ± 0.44 | 0.61 | 7.25 ± 0.80 | 7.37 ± 0.80 | 7.72 ± 0.46 | 0.19 |
| LH (IU/mL) | 5.57 ± 0.46 | 5.59 ± 0.48 | 5.68 ± 0.48 | 0.89 | 5.58 ± 0.48 | 5.52 ± 0.44 | 5.65 ± 0.44 | 0.74 |
| TSH (IU/mL) | 2.49 ± 0.31 | 2.54 ± 0.36 | 2.51 ± 0.37 | 0.88 | 2.52 ± 0.31 | 2.45 ± 0.34 | 2.44 ± 0.33 | 0.56 |

**Table 9.** ANOVA analysis studies of rs6166, rs2296545 SNPs, and PCOS subjects.

| Covariates | rs6166 | | | | rs2296545 | | | |
|---|---|---|---|---|---|---|---|---|
| | CC = 76 | CT = 16 | TT = 04 | *p*-Value | CC = 59 | CG = 25 | GG = 12 | *p*-Value |
| Age (Years) | 30.80 ± 5.87 | 31.31 ± 4.06 | 29.75 ± 6.08 | 0.89 | 30.42 ± 5.76 | 31.76 ± 5.19 | 31.00 ± 5.67 | 0.77 |
| Weight (kgs) | 73.99 ± 10.81 | 71.21 ± 12.91 | 75.95 ± 14.13 | 0.70 | 74.24 ± 10.67 | 73.76 ± 12.40 | 70.23 ± 11.82 | 0.78 |
| BMI (kg/m$^2$) | 29.30 ± 4.51 | 27.11 ± 5.51 | 31.08 ± 5.47 | 0.26 | 29.29 ± 4.46 | 29.13 ± 5.32 | 27.34 ± 5.04 | 0.70 |
| FBG (mmol/L) | 5.01 ± 0.82 | 5.36 ± 1.13 | 4.46 ± 0.69 | 0.19 | 4.95 ± 0.87 | 4.98 ± 0.69 | 5.61 ± 1.15 | 0.33 |
| FI (µIU/mL) | 10.90 ± 5.84 | 13.47 ± 8.73 | 8.93 ± 3.01 | 0.35 | 11.16 ± 6.24 | 10.42 ± 5.68 | 13.42 ± 8.18 | 0.71 |
| Creatinine (mcmol/L) | 52.70 ± 10.74 | 57.75 ± 21.85 | 55.01 ± 15.60 | 0.46 | 54.69 ± 14.64 | 50.64 ± 11.24 | 54.67 ± 10.11 | 0.69 |
| FSH (IU/mL) | 6.85 ± 2.71 | 6.93 ± 3.38 | 7.88 ± 4.23 | 0.77 | 7.24 ± 3.43 | 6.40 ± 1.51 | 6.36 ± 1.53 | 0.66 |
| LH (IU/mL) | 7.63 ± 5.12 | 7.74 ± 3.23 | 5.13 ± 2.81 | 0.60 | 8.38 ± 5.22 | 6.67 ± 4.14 | 5.25 ± 2.25 | 0.31 |
| TSH (IU/mL) | 2.24 ± 0.87 | 1.96 ± 0.91 | 2.34 ± 0.56 | 0.61 | 2.23 ± 0.74 | 2.22 ± 1.12 | 2.02 ± 0.90 | 0.87 |
| Testosterone (nmol/L) | 1.80 ± 0.81 | 2.29 ± 1.12 | 2.14 ± 1.54 | 0.20 | 1.83 ± 0.89 | 1.88 ± 0.65 | 2.21 ± 1.36 | 0.70 |
| TC (mmol/L) | 5.10 ± 1.08 | 5.09 ± 0.95 | 4.26 ± 1.34 | 0.31 | 5.08 ± 1.12 | 5.05 ± 0.97 | 4.96 ± 1.14 | 0.97 |
| TG (mmol/L) | 1.82 ± 1.08 | 1.80 ± 1.21 | 1.27 ± 0.58 | 0.61 | 1.93 ± 1.25 | 1.64 ± 0.68 | 1.44 ± 0.79 | 0.58 |
| HDL-c (mmol/L) | 0.63 ± 0.23 | 0.67 ± 0.27 | 0.61 ± 0.25 | 0.86 | 0.63 ± 0.24 | 0.61 ± 0.23 | 0.69 ± 0.23 | 0.85 |
| LDL-c (mmol/L) | 3.64 ± 0.91 | 3.61 ± 0.96 | 3.07 ± 0.95 | 0.48 | 3.58 ± 0.96 | 3.70 ± 0.85 | 3.62 ± 0.91 | 0.92 |

## 4. Discussion

The purpose of this study was to investigate the molecular role of rs6166 and rs2296545 SNPs in Saudi women with clinically confirmed FI and PCOS. The current study confirmed that none of the genotype or allele frequencies were associated with either FI or PCOS when FI or PCOS group was compared with the control group ($p > 0.05$). Additionally, both regression and Anova analyses confirmed no significant role of rs6166 or rs2296545 SNPs in women with FI and PCOS (p>0.05). In our study, HWE analysis showed a positive association, which was confirmed as there were no errors in genotyping.

The World Health Organization has confirmed infertility as a growing and complicated health issue among infertile couples [45]. Ovulation disorders, PCOS, Amenorrhea, uterine abnormalities, and fallopian tube blockages are considered causes of infertility in women [46]. Age, obesity, smoking, drinking, and stress are risk factors in FI [47]. A recent review confirmed that the prevalence of infertility is increasing globally, with similar contributions from both partners and endocrine systems, such as thyroid dysfunction, which should be monitored in male and female couples to improve chances of successful pregnancy development [1]. PCOS is one of the most common causes of infertility and may

contribute to secondary infertility. One of the most significant differences between PCOS and FI is the form of infertility in women with FI, whereas in women with PCOS, infertility is an endocrine condition with a wide variety of signs and symptoms, including hirsutism, obesity, acne, oligo/amenorrhea, and subfertility. PCOS is a curable cause of infertility, but primary or secondary infertility is not curable and has restricted alternatives for conception in women with FI. Presently, PCOS in reproductive-aged women is considered a common endocrine, neuroendocrine, and metabolic disorder, which is characterized by polycystic ovaries, ovulatory dysfunction, and hyperandrogenism, and the global prevalence of PCOS is confirmed to be at 6–10% [48]. PCOS is very common in premenopausal women, and patients with PCOS have ovarian dysfunction and hyperandrogenism. The irregularity of menstruation in women leads to anovulatory infertility, which indicates that women with PCOS become pregnant and tend to experience miscarriage or premature delivery as pregnancy complications [49]. Women with PCOS have an 11-fold increased prevalence of metabolic syndrome, which can be further aggravated, eventually leading to severe CVD. Oligomenorrhea or irregular menstrual bleeding is another contributor to infertility in women with PCOS and can progress to endometrial or uterine cancer [50]. The susceptibility loci for PCOS include FSHR, LH/choriogonadotropin receptor, THADA, and DENNDIA [51]. FSH is a member of the glycoprotein hormone family, which also includes LH and TSH. FSH is considered an essential hormone for gamete production during the reproductive period, and gonadal development and maturation during puberty. Both mutations and SNPs in *FSHR* have been shown to impair FSH signaling, resulting in infertility. Splicing variants or SNPs in *FSHR* gene were found to have little impact on infertility [52]. In our study, the rs6166 SNP was studied in both FI and PCOS, and showed a negative association. Additionally, serum levels of FSH, LH, and TSH were normal in women with FI and PCOS [53,54]. Global studies carried out between the rs6166 SNP and FI [55–60] as well as PCOS [61–66] showed all forms of association. Different ethnicities have documented varied analyses of the serum levels of FSH and the rs6166 SNP in *FSHR*. This may be due to the influence of environmental factors in specific ethnicities. Some meta-analysis studies documented the association between rs6166 SNP and different diseases in women [36,67–70]. A reason for choosing the rs2296545 SNP in *RNLS* was to validate its role in the Saudi population. The role of 2296545 SNP has been studied in T2DM, HTN, stroke, chronic kidney disease, coronary artery disease [39] infertility [38], and other human diseases [43]. In Saudi Arabia, the prevalence of diabetes and obesity is increasing, and the family history of chronic diseases is increasing in Saudi families [22,71,72]. In this study, the rs229645 SNP was studied in both women with FI and PCOS, and the results confirmed a negative association; however, the role of the missense variant (GG genotype as Asp37Glu) was found to be highly documented in women with PCOS. Another study performed in Pakistani women showed a positive association between infertility and rs2296545 SNP [38]. The rs2296545 SNP is strongly associated with other human diseases globally [39,43]. Meta-analysis studies on HTN showed a positive association with the rs2296545 SNP in *RNLS* gene [73–75].

Missense variants can alter protein-coding sequences resulting in amino acid substitutions. Most loss-of-function variations were dispersed throughout the entire protein, whereas pathogenic missense variants tended to be concentrated within particular domains or sections of the encoded proteins. The role of missense variants in a healthy population leads to the negative selection of protein structure and function, which is reflected in the functional positions of proteins [76,77]. In our study, we documented 2.1% of the rs6166 SNP variants in control participants and 10.4% of the rs2296545 SNP variants (Table 4). However, the results of our genotyping analysis showed a negative association with both SNPs in patients with FI and PCOS. Previously, the database contained over 100 million validated SNPs. Missense variants at the protein level can affect protein flexibility, chemical interactions, protein function, and enzyme activity. The important role of SNPs in missense variants can lead to abnormal protein localization in human cells by inactivating targeting signals [78]. This study established the role of missense variants in women's diseases,

specifically FI and PCOS, and documented 4.2% of missense variants (TT genotypes) in FI and PCOS patients with rs6166 SNP variants. Missense variants in the rs2296545 SNP were found in 12.5% of women with PCOS and 4.2% of women with FI. Overall, missense variations had a significant impact on all three groups studied (FI, PCOS, and control subjects), and the overall results validated this negative association. Age was one of the most important factors in our study. When compared to the age of women with FI (30.38 ± 5.39) and PCOS (30.84 ± 5.58), the mean age of control women (31.39 ± 6.70) was found to be higher, and the influence of age leads to enhanced levels of weight, which in turn affects BMI levels. In our study, control women were found to be obese (30.68 ± 4.53), whereas women with FI (29.37 ± 4.44) and PCOS (29.01 ± 4.76) were found to be overweight. Physical inactivity levels currently pose a health burden on Saudi females, perhaps leading to a higher incidence of obesity or BMI compared to males [72,79,80]. A previous study in Saudi Arabia found that 18.2% of adolescents were overweight, 11.5% were obese, and 54.5% of obese female adolescents came from families that earned more than 10,000 SR per month. The findings of this study indicate that dietary patterns are innutritious, indicating the consumption of high-calorie meals deficient in iron and fiber [81]. Figure 3 depicts the classification of BMI levels among the three groups participating in the current study; 59.4% of the controls were obese, whereas 49% and 40.6% of the FI and PCOS groups were obese, respectively (Table 10).

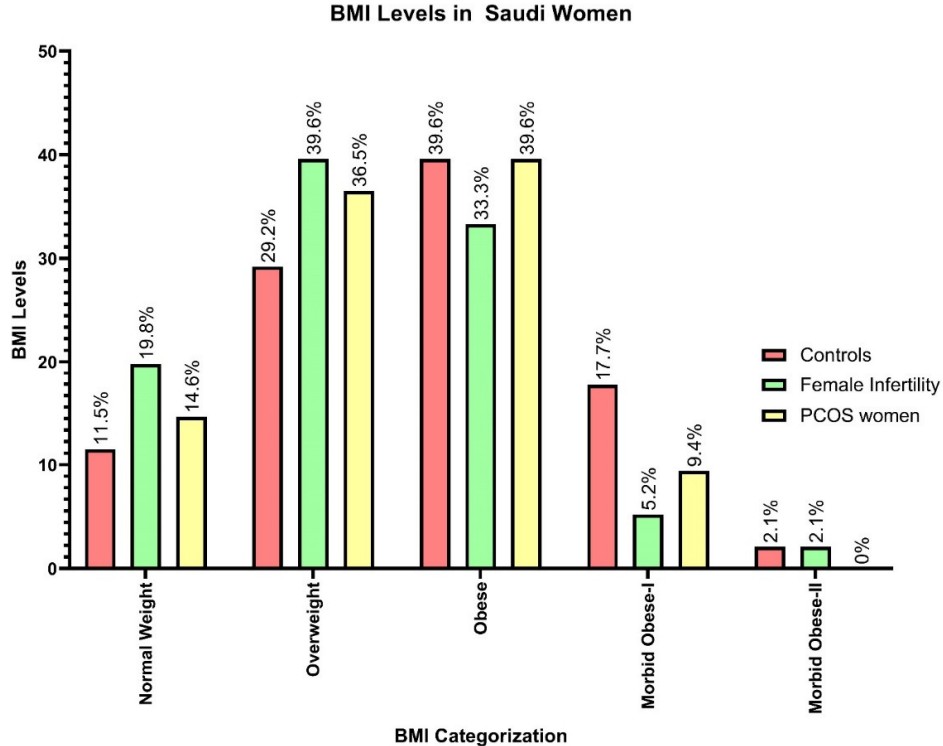

**Figure 3.** Categorization of BMI levels in FI, PCOS, and control women involved in this study.

**Table 10.** Classification of BMI in the three groups of studied women.

| BMI Categorization | Controls (n = 96) | FI (n = 96) | PCOS (n = 96) |
|---|---|---|---|
| Normal | 11 (11.5%) | 19 (19.8%) | 14 (14.6%) |
| Overweight | 28 (29.2%) | 38 (39.6%) | 35 (36.5%) |
| Obese | 38 (39.6%) | 32 (33.3%) | 38 (39.6%) |
| Morbid obese I | 17 (17.7%) | 05 (5.2%) | 09 (9.4%) |
| Morbid obese II | 02 (2.1%) | 02 (2.1%) | 00 (0.0%) |

Family history was defined as the documented medical history of biological and blood-related family members. Globally, diabetes and HTN are common family histories in many families. Diabetes can occur because of a combination of physical inactivity and family history of diabetes. Familial diabetes can lead to the development of HTN, and familial HTN can lead to the development of HTN. BMI is a comprehensive marker for the development of HTN [82,83]. HTN and diabetes commonly coexist, and are major risk factors for the development of CVD. However, both diabetes and HTN are heritable traits influenced by multiple genetic factors [84]. Saudi Arabia has a consanguinity rate of 42–67%. Evidence indicates HTN, diabetes, and CVD are associated with consanguinity [85]. Consanguineous marriages document a 50% risk from the maternal side and 20% risk from the paternal side, and the prevalence of T2DM increases in consanguineous marriages in Saudi Arabia. In our study, we documented the details of family history, apart from FI and PCOS, among control subjects and women with PCOS. However, for the FI cases, we did not record their family history of diseases other than FI, which is a limitation of our study. Figure 4 displays the prevalence of family histories in controls and women with PCOS. The prevalence of a family history of diabetes was 15.6% and 16.7%, and HNT was 3.1% and 6.3% in the control and PCOS groups, respectively and of diabetes and HTN combined was 10.4% in both groups. Our study may predict that healthy controls will develop chronic diseases in the future based on their history and family history of consanguinity; however, the recruitment age for inclusion criteria for healthy controls was determined to be 18–40 years of age.

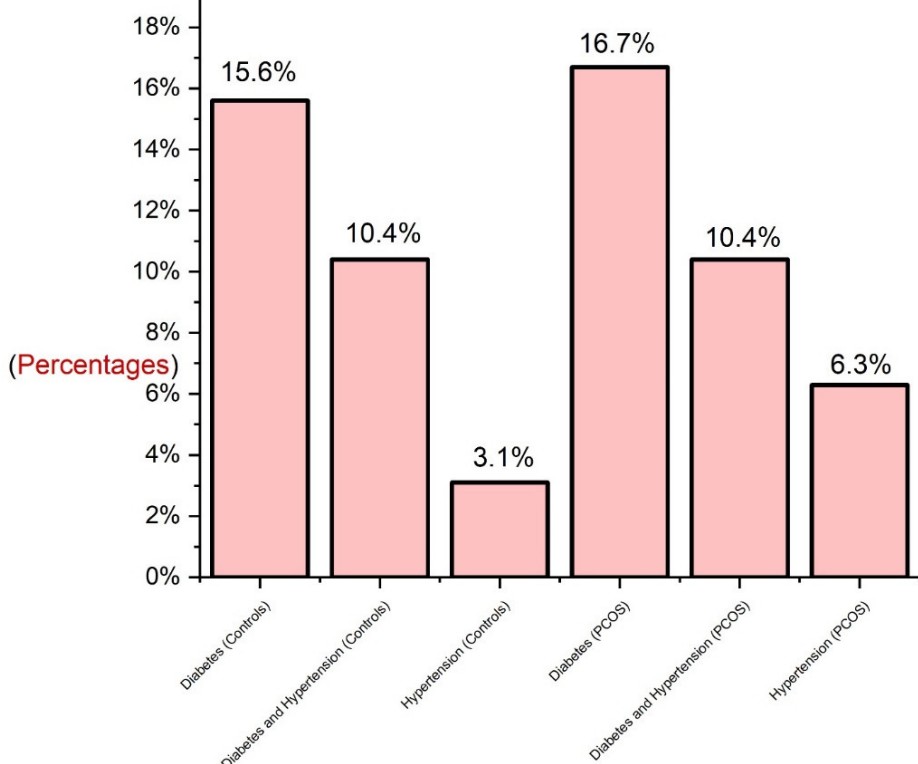

**Figure 4.** Prevalence of family histories of diabetes and hypertension in PCOS and control women.

The strength of this study lies in the recruitment of Saudi women, Sanger sequencing analysis, and the study of FI and PCOS. The lack of other family history details in women with FI apart from infertility, not documenting consanguinity details, and screening the single SNPs in the *FSHR* and *RNLS* genes were the limitations of this study. The other limitations of this study could be the lower sample size and not recording the details of pregnancy failures.

## 5. Conclusions

The results of our study confirmed that none of the genotypes or allele frequencies of rs6166 and rs2296545 SNPs in *FSHR* and *RNLS* genes were associated with FI and PCOS in Saudi women. None of the statistical analyses, such as the regression model or ANOVA, showed a significant association. Future studies should investigate the effects of rs6166 and rs2296545 SNPs on the expression of *FSHR* and *RNLS* genes along with other SNPs present in *FSHR* and *RNLS*. Future studies with well-designed, global ethnicities, and large sample sizes are needed to validate the current study findings. This study attempted to explain the reason for receiving negative impacts of both SNPs in the Saudi population.

**Author Contributions:** Conceptualization, A.F.A. and I.A.K.; methodology, S.F.A., A.A.A. and I.A.K.; software, F.M.A., R.F. and R.A.; validation, S.A., R.F., N.A.A. and I.A.K.; formal analysis, A.F.A., F.M.A. and I.A.K.; investigation, A.F.A. and I.A.K.; resources, M.A.A. and J.A.-M.; data curation, A.F.A. and I.A.K.; writing—original draft preparation, A.F.A. and I.A.K.; writing—review and editing, A.F.A. and I.A.K.; visualization, A.F.A. and I.A.K.; supervision, A.F.A. and I.A.K.; project administration, A.F.A. and I.A.K.; funding acquisition, A.F.A. and I.A.K. All authors have read and agreed to the published version of the manuscript.

**Funding:** The authors extend their appreciation to the Deputyship for Researchers and Innovation, the "Ministry of Education" in Saudi Arabia for funding this research (IFKSUOR3-126-1).

**Institutional Review Board Statement:** The Institutional Review Board at King Saud University approved the ethical grant for female infertility (E-19-4344) and PCOS (E-20-5339) projects in the premises of the College of Medicine.

**Informed Consent Statement:** All the women (n = 288) who participated in this study signed a patient consent form.

**Data Availability Statement:** Data are not applicable in this study.

**Acknowledgments:** The authors extend their appreciation to the deputyship for Research and Innovation, "Ministry of Education" in Saudi Arabia for funding this research (IFKSUOR3-126-1)": IFKSUOR3-126-1.

**Conflicts of Interest:** There is no conflict of interest towards this study.

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
