# Peer review of "Molecular Role of Asn680Ser and Asp37Glu Missense Variants in Saudi Women with Female Infertility and Polycystic Ovarian Syndrome"

_cimb, doi:10.3390/cimb45070348_

Round 1
Reviewer 1 Report
Comments and Suggestions for Authors
Comments about the manuscript:
“Molecular role of Asn680Ser and Asp37Glu missense variants in Saudi women with female infertility and polycystic ovarian syndrome”
Polycystic ovary syndrome is a common cause of female infertility associated with the renalase gene (RNLS). Single nucleotide polymorphisms (SNPs) rs6166 and rs2296545 have not yet been identified in Saudi women with infertility and polycystic ovary syndrome. The aim of this study was to investigate the role of these two SNPs in the FSH (FSHR) and renalase (RNLS) receptor genes in these individuals. For this, DNA was isolated from the blood of 96 healthy controls, 96 patients with infertility and 96 patients with polycystic ovary syndrome and PCRs were performed from SNPs rs6166 and rs2296545. After statistical analysis of the results, rs6166 and rs2296545 did not appear to be involved in infertility or polycystic ovary syndrome.
This work provides useful elements given the size of the samples (controls, women with infertility and polycystic ovary syndrome). It is, however, a complex work with many elements and which I found difficult to read. It is a complex work given the variations in the compositions of the three groups: women from control group are on average older than those of the other two groups.
It seems that the point that needs to be completed is precisely the composition of the three groups. It would seem useful to divide the total number of individuals into several age categories (left to the discretion of the authors according to the data available) in order to have a more precise distribution.
Here are some remarks.
Page 4, lines 139-141. “The first group consisted of infertile women, the second consisted of Saudi women diagnosed with PCOS, and the final group consisted of healthy controls”: Given that polycystic ovary syndrome is a factor of infertility, does the group called IF also include women known to have this syndrome or are they different cases of infertility (if we can know)? This point is not very clear to me.
Page 4, line 161: “Women who participated in this study (n=288) were in the age range of 18-40 years.” It would be useful to know the distribution of age groups in the different study groups. For this, it would be useful to define categories (see general remark).
Page 4, lines 177-178 “In this study, we collected 6 ml of peripheral blood from 192 Saudi women (96 patients with PCOS and 96 healthy controls)”: how many women of different age categories in these groups? (Same question as above).
Page 6, line 234: table 1 is not called in the text. What data are shown in columns 2 and 3: averages? Please explain.
Page 7, line 236, table 2. Same remark as for table 1: what data are shown in columns 2 and 3: averages? Please explain.
Page 11, line 262. “The mean age,”: it would be interesting to have the distribution of ages according to categories.
Page 14, line 379 write “amenorrhea” (small letter) instead of “Amenorrhea (small letter)”.
Author Response
Reviewer comments
Dear Reviewer,
We as a team appreciate your time and effort for spending towards our manuscript. Your comments have made us to improve the manuscript and we deeply thankful to you. We have carefully justified all your comments and incorporated in the revised manuscript. Please find the justification below and we hope, we have justified all the quires.
Reviewer 1
Title: “Molecular role of Asn680Ser and Asp37Glu missense variants in Saudi women with female infertility and polycystic ovarian syndrome”
Polycystic ovary syndrome is a common cause of female infertility associated with the renalase gene (RNLS). Single nucleotide polymorphisms (SNPs) rs6166 and rs2296545 have not yet been identified in Saudi women with infertility and polycystic ovary syndrome. The aim of this study was to investigate the role of these two SNPs in the FSH (FSHR) and renalase (RNLS) receptor genes in these individuals. For this, DNA was isolated from the blood of 96 healthy controls, 96 patients with infertility and 96 patients with polycystic ovary syndrome and PCRs were performed from SNPs rs6166 and rs2296545. After statistical analysis of the results, rs6166 and rs2296545 did not appear to be involved in infertility or polycystic ovary syndrome.
- A) This is the concept and output of this study.
This work provides useful elements given the size of the samples (controls, women with infertility and polycystic ovary syndrome). It is, however, a complex work with many elements and which I found difficult to read. It is a complex work given the variations in the compositions of the three groups: women from control group are on average older than those of the other two groups.
- A) This work was carried out with a good sample size and it includes three groups of subjects. This manuscript was edited with native experts. Unfortunately, control women were found to be elder than cases because both female infertility and PCOS women were selected during the reproductive age.
It seems that the point that needs to be completed is precisely the composition of the three groups. It would seem useful to divide the total number of individuals into several age categories (left to the discretion of the authors according to the data available) in order to have a more precise distribution.
- A) Based on your comment, we have made a table and defined the clinical details and genotype details into 3 groups (i) between 18-20 years, (ii) between 21-30 years and (iii) between 31-40 years of age.
Here are some remarks.
Page 4, lines 139-141. “The first group consisted of infertile women, the second consisted of Saudi women diagnosed with PCOS, and the final group consisted of healthy controls”: Given that polycystic ovary syndrome is a factor of infertility, does the group called IF also include women known to have this syndrome or are they different cases of infertility (if we can know)? This point is not very clear to me.
- A) In this study, we have enrolled (i) PCOS women, (ii) Female infertility (FI) women and (iii) control women. PCOS is a factor of infertility but we have enrolled two different (PCOS and FI) groups of cases along with a matching control.
Page 4, line 161: “Women who participated in this study (n=288) were in the age range of 18-40 years.” It would be useful to know the distribution of age groups in the different study groups. For this, it would be useful to define categories (see general remark).
- A) It was a good suggestion and based on your comment, we have created one more table and labelled as (Table#3) to distribute the women as per the 3 groups of ages (1= 18-20 years, 2=21-30 years and 3=31-40 years).
Page 4, lines 177-178 “In this study, we collected 6 ml of peripheral blood from 192 Saudi women (96 patients with PCOS and 96 healthy controls)”: how many women of different age categories in these groups? (Same question as above).
- A) In this study, (i) 96 PCOS women, (ii) 96 Female Infertility women and (iii) 96 healthy controls were selected for this study. All the women who have participated in this study were in between 18-40 years of age.
Page 6, line 234: table 1 is not called in the text. What data are shown in columns 2 and 3: averages? Please explain.
- A) Table 1 contains clinical data values for female infertility cases (column 2) and control participants (column 3), given with mean ± standard deviation. Table-1 is now described in the main document under the results sub-section 3.2
Page 7, line 236, table 2. Same remark as for table 1: what data are shown in columns 2 and 3: averages? Please explain.
- A) Table 2 contains clinical data values for PCOS cases (column 2) and control participants (column 3), given with mean standard deviation. Table-2 is now described in the main document under the results sub-section 3.3.
Page 11, line 262. “The mean age,”: it would be interesting to have the distribution of ages according to categories.
- A) We have created one more table (Table #3) and mentioned in detail as per your suggestion.
Page 14, line 379 write “amenorrhea” (small letter) instead of “Amenorrhea (small letter)”.
- A) Now, we have updated in the revised manuscript.
Reviewer 2 Report
Comments and Suggestions for Authors
The manuscript “Molecular role of Asn680Ser and Asp37Glu missense variants in Saudi women with female infertility and polycystic ovarian syndrome” contributed to giving information about the relationship between SNPs and female infertility risk in Saudi women. Personally, I am sure that the accumulation of information such as this manuscript is very important for future clinical applications. Although there are some limitations, I am wondering how can it be improved for future research. Also, it needs additional schemes of their results to make it easier for the reader to read.
Some points have to be corrected.
Major points
1. Unfortunately, both rs6166 and rs2296545 showed no role in FI and PCOS in Saudi women. The main reason may be the use of elderly women as control. Otherwise, other factors are Saudi-specific ethnicities such as increasing obesity and consanguineous marriage. However, increasing obesity is a global problem in the world. Describe the authors’ ideas about the reasons for no association between them in the discussion concretely.
2. It needs to describe the reason why the authors focused on rs2296545 SNPs. According to the founding in Pakistan (Fatima et al., 2019), the authors focused on Renalase (rs2576178 and rs10887800) to examine the association between SNPs and causes of infertility.
3. Figures and Tables needs to rearrange. Properly, adding them to text makes it easier to read.
3. In this study, the authors focused on the association between SNPs (rs6166 and rs2296545) and female infertility. Although there is no significant association, It is better to add a schematic model of their relationship and how they are related to pregnancy failures.
Minor points
1. Line 89: Add “-” between “family” and “based”.
2. Lines 99-100 and 108-109: Check the abbreviation of FSHR.
3. Line 117: Add “of” before “the above”.
4. Lines 119 and 125: Check the abbreviation of Renalase. In line 115, the abbreviation of renalase was described as RNLS.
5. Line 290: Amend “are” to “is”.
6. Line 303: Please check “found be” in the sentence.
7. Line 304: Amend “signiifcant” to “significant”.
8. Line 317: Remove “the” in “all the three groups”.
9. Line 330: Remove “the” in “the Table-5”.
10. Line 413: Amend “in” to “on”.
Comments on the Quality of English Language
Minor editing of English language required
Author Response
Dear Reviewer,
We as a team appreciate your time and effort for spending towards our manuscript. Your comments have made us to improve the manuscript and we deeply thankful to you. We have carefully justified all your comments and incorporated in the revised manuscript. Please find the justification below and we hope, we have justified all the quires.
Reviewer comments 2
The manuscript “Molecular role of Asn680Ser and Asp37Glu missense variants in Saudi women with female infertility and polycystic ovarian syndrome” contributed to giving information about the relationship between SNPs and female infertility risk in Saudi women. Personally, I am sure that the accumulation of information such as this manuscript is very important for future clinical applications. Although there are some limitations, I am wondering how can it be improved for future research. Also, it needs additional schemes of their results to make it easier for the reader to read.
- A) Thank you very much for your comment. The prevalence of PCOS and infertility, specifically female infertility is increasing in the Kingdom of Saudi Arabia. Based on design of this study, we have enrolled a couple of markers such as Asn680Ser and Asp37Glu, which were found to be missense variants. We will try to justify all the raised comments.
Some points have to be corrected.
Major points
- Unfortunately, both rs6166 and rs2296545 showed no role in FI and PCOS in Saudi women. The main reason may be the use of elderly women as control. Otherwise, other factors are Saudi-specific ethnicities such as increasing obesity and consanguineous marriage. However, increasing obesity is a global problem in the world. Describe the authors’ ideas about the reasons for no association between them in the discussion concretely.
- A) One of the major reasons for not associating in this study is due to obesity, though the prevalence of obesity is increasing throughout the world. The prevalence of obesity is high in Saudi women in comparison with Saudi men. We have tried to select the matching controls and cases towards this study. The other possibility is low sample size, which was described in the limitations of this study.
- It needs to describe the reason why the authors focused on rs2296545 SNPs. According to the founding in Pakistan (Fatima et al., 2019), the authors focused on Renalase (rs2576178 and rs10887800) to examine the association between SNPs and causes of infertility.
- A) The renalase gene is closely associated with female infertility, type 2 diabetes, hypertension and many other diseases. The prevalence of these diseases is increasing in Saudi Arabia and on a common basis we have selected rs2296545 SNP in renalase gene to study in Saudi women confirmed with PCOS and female infertility women. This was the main purpose of selecting rs2296545 SNP in renalase gene.
- Figures and Tables needs to rearrange. Properly, adding them to text makes it easier to read.
- A) This was amended by CIMB journal format using MS word template. I have tried my best in arranging but after the submission, the CIMB system will amend as per their convenience
- In this study, the authors focused on the association between SNPs (rs6166 and rs2296545) and female infertility. Although there is no significant association, It is better to add a schematic model of their relationship and how they are related to pregnancy failures.
- A) Unfortunately, there was no association but a graphical model was presented. The personnel details of pregnancy failures were not recorded which could be the other limitations of this study. We have described in the revised manuscript under limitations of this study and highlighted with yellow color.
Minor points
- Line 89: Add “-” between “family” and “based”.
- A) Now, we have added.
- Lines 99-100 and 108-109: Check the abbreviation of FSHR.
- A) Thank you very much for your comment. We have updated in the revised manuscript.
- Line 117: Add “of” before “the above”.
- A) We have added in the revision.
- Lines 119 and 125: Check the abbreviation of Renalase. In line 115, the abbreviation of renalase was described as RNLS.
- A) Once again thank you for your comment. We have rectified the error.
- Line 290: Amend “are” to “is”.
- A) We have updated now.
- Line 303: Please check “found be” in the sentence.
- A) We have updated now.
- Line 304: Amend “signiifcant” to “significant”.
- A) We have updated now.
- Line 317: Remove “the” in “all the three groups”.
- A) We have removed now in the revised manuscript.
- Line 330: Remove “the” in “the Table-5”.
- A) We have removed the word “the” in the revised manuscript.
- Line 413: Amend “in” to “on”.
- A) We have amended in with on in the revised manuscript.

Round 2
Reviewer 2 Report
Comments and Suggestions for Authors
I think that the revised manuscript has been fundamentally improved and that it includes the contents requested by the referees and editorial team.